# Autotrophic and Heterotrophic Growth Conditions Modify Biomolecole Production in the Microalga *Galdieria sulphuraria* (Cyanidiophyceae, Rhodophyta)

**DOI:** 10.3390/md18030169

**Published:** 2020-03-18

**Authors:** Roberto Barone, Lorenzo De Napoli, Luciano Mayol, Marina Paolucci, Maria Grazia Volpe, Luigi D’Elia, Antonino Pollio, Marco Guida, Edvige Gambino, Federica Carraturo, Roberta Marra, Francesco Vinale, Sheridan Lois Woo, Matteo Lorito

**Affiliations:** 1Department of Agricultural Science, University of Naples Federico II, Via Università, Portici, 80138 Naples, Italy; robmarra@unina.it (R.M.); matteo.lorito@unina.it (M.L.); 2Department of Pharmacy, University of Naples Federico II, Via D. Montesano, 80138 Naples, Italy; lorenzo.denapoli@unina.it (L.D.N.); luciano.mayol@unina.it (L.M.); 3Department of Science and Technologies (DST), University of Sannio, 82100 Benevento, Italy; 4Institute of Food Sciences, National Research Council (ISA-CNR), Via Roma 64, 83100 Avellino, Italy; mgvolpe@isa.cnr.it; 5Department of Biology, University of Naples Federico II, Via Cinthia, 80138 Naples, Italyantonino.pollio@unina.it (A.P.); ed.gambino@gmail.com (E.G.); federica.carraturo@unina.it (F.C.); 6Department of Veterinary Medicine and Animal Production, University of Naples Federico II, Via Federico Delpino, 80138 Naples, Italy; frvinale@unina.it; 7IPSP-CNR-Via Università-Portici, 80138 Naples, Italy

**Keywords:** *Galdieria sulphuraria*, microalga, fungi, autothrophy, heterotrophy, fatty acids, ATR-FTIR

## Abstract

Algae have multiple similarities with fungi, with both belonging to the *Thallophyte*, a polyphyletic group of non-mobile organisms grouped together on the basis of similar characteristics, but not sharing a common ancestor. The main difference between algae and fungi is noted in their metabolism. In fact, although algae have chlorophyll-bearing thalloids and are autotrophic organisms, fungi lack chlorophyll and are heterotrophic, not able to synthesize their own nutrients. However, our studies have shown that the extremophilic microalga *Galderia sulphuraria* (GS) can also grow very well in heterotrophic conditions like fungi. This study was carried out using several approaches such as scanning electron microscope (SEM), gas chromatography/mass spectrometry (GC/MS), and infrared spectrophotometry (ATR-FTIR). Results showed that the GS, strain ACUF 064, cultured in autotrophic (AGS) and heterotrophic (HGS) conditions, produced different biomolecules. In particular, when grown in HGS, the algae (i) was 30% larger, with an increase in carbon mass that was 20% greater than AGS; (ii) produced higher quantities of stearic acid, oleic acid, monounsaturated fatty acids (MUFAs), and ergosterol; (iii) produced lower quantities of fatty acid methyl esters (FAMEs) such as methyl palmytate, and methyl linoleate, saturated fatty acids (SFAs), and poyliunsaturated fatty acids (PUFAs). ATR-FTIR and principal component analysis (PCA) statistical analysis confirmed that the macromolecular content of HGS was significantly different from AGS. The ability to produce different macromolecules by changing the trophic conditions may represent an interesting strategy to induce microalgae to produce different biomolecules that can find applications in several fields such as food, feed, nutraceutical, or energy production.

## 1. Introduction

Microalgae are unicellular organisms commonly found in fresh and marine waters. They are very similar to fungi [1], both are morphologically undifferentiated and included in the group of *Thallophytes*. However, the main difference is that algae require light, contain chlorophyll, and are autotrophs. Members are characterized by a high biodiversity whose potential, in terms of the production of high value biological molecules, is yet to be explored and exploited [2]. Microalgae cultivation can provide diverse essential nutrients, including carbohydrates, proteins, and lipids, as well as pigments, vitamins, bioactive compounds, and antioxidants [3,4]; substances that can be utilized in nutraceuticals, pharmaceuticals, biofuels, health supplements, and the cosmetic industry. Furthermore, microalgae cultivation provides a potential strategy to produce an alternative food source for both humans and animals. This feature, plus the ability of microalgae to grow more rapidly than vascular plants, satisfies the need for large-scale, cost-effective, high nutritional value production [5]. Therefore, microalgae represent an interesting resource in the biotechnology field, as they are able to quickly reach a high level of biomass and produce a large quantity of fatty acids (FAs) such as palmitic acid (C16:0), myristic acid (C14:0), monounsatured (MUFAs), polyunsatured FAs (PUFAs), and fatty acid methyl esters (FAMEs), molecules extremely interesting for commercial applications. Microalgae also produce pharmacologically active molecules with immunomodulatory, anti-inflammatory, antihypercholesterolemic, antioxidant, anticancer, and antidiabetic properties [6,7,8]. The metabolic flexibility of microalgae allows them to grow in both autotrophic and heterotrophic conditions [9]. The benefit trade-offs are diverse for algae, whereby the autotrophic growing condition is preferred from an efficiency point of view, but it provides a limited growth of biomass, whereas the biomass obtained under heterotrophic growing conditions is greater, but requires additional external carbon sources that are energetically expensive [10,11]. The heterotrophic cultivation of *Chlorella vulgaris*, the oldest microalgae exploited for commercial application, has demonstrated higher biomass yields than the autotrophic cultivation, with higher lipid productivity [12]. 

*Galdieria sulphuraria* (GS; Cyanidiophyceae, Rhodophyta) is an ancient extremophilic unicellular red microalga capable of growing in hot springs at low pH [13,14] all around the world. It shows optimal growth conditions at pH 1.5 and temperatures in the range of 35–45 °C,extreme growth conditions that prevent bacterial contamination, one of the major problems faced with large scale microalgae cultivation [15,16]. Moreover, GS is able to grow photoautotrophically, heterotrophically, and mixotrophically, but to date, not much is known about the morphological and biochemical changes induced by the different growing conditions or the effect on the production of different biomolecules by the microalgae. It has been noted that heterotrophic growth of GS leads to cytological changes in the cell size, probably due to reduced chloroplast size and increased number of mitochondria, the organelles directly connected with nutrition [17]. GS exhibits a high metabolic flexibility that is matched by few other microorganisms, demonstrating the ability to thrive on more than 50 different carbon sources such as sugars, sugar alcohols, tricarboxylic-acid-cycle intermediates, and amino acids [18,19,20,21]. In addition, this genus has very high daily productivity of various bioactive compounds [15] and significant potential as a source of antioxidants and macronutrients, features that have driven interest towards conduct investigations on this Cyanidiophycea for its potential biotechnological applications [22,23,24,25]. In the present study, a comparison was made on the growth and metabolism of GS cultured under both autotrophic and heterotrophic conditions, and the different biomolecules obtained under the different growing conditions were characterized and identified by using a combination of techniques: scanning electron microscope (SEM), gas chromatography/mass spectrometry (GC/MS), and infrared spectrophotometry (ATR-FTIR). The well-known, studied, and commercialized microalgae *Spirulina platensis* (Sp) was grown in autotrophic conditions and used in this study as a comparison species. The final aim of this investigation was to verify the possibility of directing or manipulating the metabolic flexibility of GS as a tool to induce the production of biomass and biomolecules that can be of interest to several important fields such as food, feed, nutraceutical, or energy production industries. 

## 2. Results

### 2.1. Scanning Electron Microscopy

An increase in the cell dimension of GS grown in heterotrophic conditions with respect to the autotrophic conditions was detected by SEM analysis. The average cell size of heterotrophic (HGS) conditions was about 30% bigger than autotrophic (AGS) conditions (Figure 1). Moreover, AGS showed different element contents with respect to the heterotrophic conditions (Figure 2). 

Area values are reported in Table 1. A significant increase in carbon mass of 20% for the heterotrophic growth conditions was observed. 

### 2.2. GC-MS Analysis

Fatty acid (FA) composition of *Galdieria sulphuraria* strain ACUF 064 cultivated in autotrophic (AGS) and heterotrophic (HGS) conditions and *Spirulina platensis* (Sp) for comparison are reported in Table 2. 

In the autotrophic conditions, higher levels of fatty acid methyl esters (FAMEs) were present, especially methyl palmitate and methyl linoleate. Another compound present in higher quantities in AGS was phytol (PYT), an acyclic diterpenoid alcohol constituent of chlorophyll. The heterotrophic condition influenced the production of ergosterol, a phytosterol, stearic acid (STA) and oleic acid, present in higher concentrations with respect to the autotrophic condition. Omega 3 long chain FAs, such as EPA, DHA, and arachidonic acid, were found neither in the autotrophic nor in the heterotrophic conditions. 

### 2.3. ATR-FTIR

Mean FTIR spectra of GS strain ACUF 064 cultivated in autotrophic (AGS) and heterotrophic (HGS) conditions and Sp are shown in Figure 3. Each spectrum is the average of three raw spectra originating from five samples. 

Each FTIR spectrum is formed by peaks arising from the infrared absorption of functional groups. The vibration intensity, reported as absorbance, is proportional to the relative abundance of organic molecules such as carbohydrates, lipids, and proteins. Table 3 reports FTIR peak assignments based on spectral values indicated in the current literature [26,27]. Although a certain degree of overlapping is present, macromolecules can be identified in relation to specific wavelength ranges [28]. Lipids can be identified in the range 3000–2800 and around 1740 cm^−1^, proteins in the range 3600–3000 and around 1640 and 1540 cm^−1^, and carbohydrates in the range 1174–950 cm^−1^.

The overlapping spectra reported in Figure 3 indicated how the intensity of the peaks corresponding to proteins, lipids, and carbohydrates was greater in HGS than in AGS and Sp, with the only exception of the peak around 1540 cm^−1^, ascribable to N–H stretching of proteins, which was higher in Sp, followed by AGS. HGS was richer in polysaccharides and sugars compared to AGS and Sp, as indicated by the high absorbance in the range 1174–950 cm^−1^. Polysaccharides in HGS were highlighted by the two peaks at 1148 and 1018 cm^−1^, which were missing in AGS and Sp. Special attention should be devoted to the 950–650 cm^−1^ region, also called the “fingerprint region”. In particular, HGS showed four different sharp absorption bands (931, 850, 760, 662 cm^−1^) that represent a characteristic fingerprint of HGS, different from AGS and *Sp* that presented a very similar pattern in this region. A representative FTIR substraction spectrum of HGS minus AGS highlights the differences in the concentration of macromolecules between autotrophic and heterotrophic conditions (Figure 4A). In order to quantify the different content of macromolecules such as lipids, carbohydrates, and proteins, the second derivative of the FTIR profiles was determined (Figure 4B).

To make a quantitative determination, the integration of the second derivative peaks was carried out according to Equation (2), reported in the Materials and Methods section. HGS compared to AGS showed a greater content of proteins, lipids, and carbohydrates of 91%, 57%, and 98%, respectively. The areas are reported in Table 4.

FTIR spectra of HGS, AGS, and Sp were quite complex and required a multivariate statistical analysis for the data comparison. In this study, we used a chemometric approach based on the principal component analysis to analyze the whole spectral range and sub-ranges corresponding to specific macromolecules as reported in Table 5. Data interpretation by means of SIMCA (soft independent modeling class algorithm) algorithm (Figure 5) confirmed the differences in macromolecules between the autotrophic and heterotrophic culture conditions. 

The significant differences between the autotrophic and the heterotrophic conditions are demonstrated by the interclass distance (ID) reported in Table 5. The ID highlights the similarities between AGS and Sp (Sp-AGS), as well as the differences between them and HGS (AGS–HGS, HGS-Sp). The higher the ID value, the greater the difference. It is reported that a distance value higher than 3 is indicative of well separated samples, and therefore belonging to different classes [29].

Figure 5 shows the 3D-PCA score plot generated by the SIMCA model. This multivariate analysis permits the visualization of the class separation among HGS, AGS, and Sp. The boundary ellipse (hyperboxes) defining each cluster represents a 95% confidence interval, and the points within each cluster represent the spectrum wavelengths of each sample in the three-dimensional space. Data analysis performed in smaller ranges of the spectrum (Table 5) revealed that there were significant differences among groups. The interclass distance clearly underlines the changes in GS as a consequence of the modification of the metabolism due to the growth conditions. In fact, although Sp and AGS are different microalgae species, both grown as autotrophs, they appeared to be extremely similar with an interclass distance ranging from a minimum of 4.94 (spectrum range 2999–2800 cm^−1^) to a maximum of 21.7 (spectrum range 1575–1478 cm^−1^), whereas HGS and AGS were found to be extremely different with a minimum inter-distance of 15.4 and a maximum of 78.3 (respectively for the intervals of 1711–1576 and 1174–950 cm^−1^) due to the diverse metabolism impost by the heterotrophic and autotrophic conditions. 

## 3. Discussion

### 3.1. Scanning Electron Microscopy

According to the SEM analysis, the average size of GS cells grown in heterotrophic condition was about 30% greater than those cells produced in the autotrophic condition. This outcome is in agreement with Stadnichuk et al. [17] who reported an increase in the cell dimension of *Galdieria partita* grown in heterotrophic conditions with respect to the autotrophic conditions. Furthermore, the authors hypothesized that the outcome could be a result of D-glucose inhibition on the photosynthetic pigment apparatus. Interestingly, our findings noted a decrease in phytol (PYT) content, a constituent of chlorophyll, in HGS that could support this theory. Moreover, AGS exhibited different element contents with respect to the heterotrophic conditions, and there was a significant increase in carbon mass of 20% in the heterotrophic growth conditions. 

### 3.2. GC-MS Analysis

AGS showed a different FA composition with respect to the HGS, whereby in the autotrophic conditions, higher levels of fatty acid methyl esters (FAMEs) were present, especially for methyl palmitate, methyl linoleate, and hexadecanoic acid methyl ester with respect to the heterotrophic conditions. This outcome is quite interesting because it indicates the avoidance of the expensive phase of esterification that is necessary for the production of FAMEs for their final application as biodiesel [30,31,32]. 

Another interesting compound present in higher quantities in AGS is phytol (PYT), an acyclic diterpenoid alcohol. Its presence is most likely related to the chlorophyll in the autotrophic form. PYT and its derivatives have a vast array of actions ranging from antimicrobial, anticancer, anti-inflammatory, and immune stimulant activities, to being a hair growth facilitator [33]. Furthermore, PYT is used as a precursor for the production of synthetic forms of vitamin E [34] and vitamin K [35], and therefore of great interest in pharmaceutical applications.

The condition of heterotrophy induced GS to produce higher levels of ergosterol, as observed in fungi [36] or phytosterol in plants, with many beneficial health effects for humans, including immunomodulatory, anti-inflammatory, neuromodulatory, antihypercholesterolemic, antioxidant, anticancer, and antidiabetic properties [37]. Ergosterol is also a biological precursor of vitamin D2 (ergocalciferol) [38], and exposure to ultraviolet light causes a photochemical reaction that activates the conversion of ergosterol to ergocalciferol. In addition, ergosterol is of great importance because it undergoes photolysis when exposed to UV light (280–320 nm) to yield provitamin D2 as one of the main products, which under thermal rearrangement, is spontaneously transformed into vitamin D2 [39]. Ergosterol and derivatives have shown a wide range of health-promoting properties, such as antioxidant, anti-inflammatory, and antihyperlipidemic activities [40]. Treatments with ergosterol were able to significantly inhibit the proliferation of human epithelial type 2 (HEp-2) cells, a cell line originating from human laryngeal carcinoma, and the ergosterol derivatives were known to be a source of new potential antitumor or anti-angiogenesis chemotherapy agents [41]. Moreover, ergosterol derivatives have the ability to suppress lipopolysaccharide (LPS)-induced inflammatory responses of macrophages in vitro through the inhibition of highly proinflammatory cytokine (TNF-α) and cyclooxygenase-2 (COX-2) expression, as well as having a cytostatic effect on human colorectal adenocarcinoma cells [42]. Therefore, this molecule has promising multiple beneficial applications in the pharmacological field.

The heterotrophic condition was also found to influence the production of oleic acid, which was present in higher concentrations in comparison to the autotrophic condition. This is likely related to the increased cell dimensions of GS when cultured under heterotrophic conditions, and to the absence of chlorophyll a and phycocyanobilin biosynthesis, as previously observed by Stadnichuk et al. [17]. Oleic acid is a MUFA that finds interesting applications in the field of nutrition because it has the ability to reduce low density lipoprotein cholesterol (LDL-cholesterol), and at the same time, to promote high density lipoprotein cholesterol (HDL-cholesterol) [43,44]. Although the production of the omega 3 long chain FAs, such as EPA, DHA, and arachidonic acid, fundamental constituents of the human and animal diet [45], have not been found either under autotrophic or heterotrophic conditions, it is interesting to note that the autotrophic condition is accompanied by a general increase in PUFA, and in particular in linoleic and linolenic acid, which respectively belong to the omega 6 and omega 3 series. This outcome may have important consequences in the field of animal nutrition, in particular for freshwater fish nutrition, as they are able to synthesize EPA, DHA, and arachidonic acid from linoleic and linolenic acids.

The GC-MS data also indicated the presence of a high percentage of stearic acid (STA) in HGS, whereas in Sp this SFA was found to be present in negligible quantities. Recent studies have shown that stearic acid has favorable effects on human health. In fact, diets in which STA has been added in high percentages were able to drastically reduce LDL-cholesterol. STA applications may thus be of great interest in the pharmacological and nutraceutical fields [46].

### 3.3. Infrared Spectrophotometry

FTIR spectra of biological samples reported the macromolecular composition on the basis of the infrared absorption of functional groups [47]. The vibration intensity, reported as absorbance, is proportional to the relative abundance of organic molecules such as carbohydrates, lipids, and proteins [48]. The FTIR spectra analysis of GS provides interesting information about the changes in the macromolecule composition induced by different growth conditions, confirming the usefulness of FTIR as a fast, non-disruptive method to identify macromolecules in microalgae [49,50]. The intensity of the peaks corresponding to proteins, lipids, and carbohydrates was greater in HGS than in AGS and Sp, with the only exception with the peak observed around 1540 cm^−1^, which was higher in Sp and AGS, and was ascribable to N–H stretching of proteins. It is worth noting that Sp had a characteristic high content of proteins, as was also reported by Rafiqul et al. [51]. HGS was richer in polysaccharides and sugars when compared to AGS and Sp., as indicated by the high absorbance in the range 1174-950 cm^−1^. Polysaccharides in HGS are highlighted by the two peaks at 1148 and 1018 cm^−1^, which were similar to peaks that were present in *Chlorella vulgaris* by [52], but not present in AGS and Sp. 

The different contents of macromolecules such as lipids, carbohydrates, and proteins, in AGS, HGS, and Sp, was confirmed by the evaluation of the second derivative of the FTIR profiles that revealed that HGS, in comparison to AGS, had a greater content of proteins, lipids, and carbohydrates at 91%, 57%, and 98%, respectively.

The significant differences between the autotrophic and the heterotrophic conditions were also demonstrated by the interclass distance (ID), whereby the ID highlights the similarities between AGS and Sp, plus their apparent differences to HGS with the higher ID values indicating a greater difference. It has been reported that a distance value higher than 3 is indicative of well-separated samples, which confirms their difference [53]. The interclass distance is able to underline the changes in GS as a consequence of the modification of the metabolism. Thus, metabolic changes, from autotrophic to heterotrophic, have relevant effects on both morphological and chemical characteristics of GS.

## 4. Materials and Methods

### 4.1. Strain and Growth Medium

*Galdieria sulphuraria* (Galdieri) Merola n. 064 was obtained from the algal collection of the Department of Biology of the University of Naples Federico II (ACUF). A preliminary screening study of 43 strains showed that the strain 064 has the lowest doubling time in autotrophic and heterotrophic conditions. Modified Allen medium [54,55] (Table 6) was used for autotrophy growth, whereas the same medium supplemented with glycerol was used for heterotrophy growth. Modified Allen medium contained NaNO_3_ as a nitrogen source. The standard concentration of the nitrate was 72 g L^−1^. H_2_SO_4_ was adopted for fine setting of the initial pH at 1.5. The medium was autoclaved for 20 min before use. 

### 4.2. Growth Conditions

For microalgae culture (*Galdieria sulphuraria* in autotrophic conditions (AGS), *Galdieria sulphuraria* in heterotrophic conditions (HGS), and *Spirulina platensis* (Sp), pre-cultures of 50 mL inoculated from a single isolate picked from a solid plate were grown in 200 mL Erlenmeyer flasks housed in a climatic chamber (Gibertini, Italy) at 37 ± 1 °C. The chamber was equipped with daylight fluorescent lamps (Philips TLD 30 W/55) set at 150 µE/m^2^ s for 24/24. After 2 weeks, the pre-cultures were used to inoculate the photobioreactors. The growth was carried out in a cylindrical bubble column photobioreactor made of glass (0.04 m ID. 0.8 m high) with a 0.9 Lworking volume [56]. Air was sparged at the photobioreactor bottom by means of a porous ceramic diffuser at a volumetric flow rate ranging between 20 and 200 nl h^−1^. Filters of 0.2 µm were used to sterilize air flow inlet and outlet. The photobioreactors were housed in a climate chamber (Solar Neon) at 37 ± 1 °C. The chamber was also equipped with fluorescent lamps (Philips TLD 30 W/55) for autotrophic conditions. Heterotrophic cultures were conducted in the dark. In order to sustain the autotrophic growth in optimal conditions in the photobioreactor for long periods, the concentration of salts in the modified Allen culture medium was doubled with respect to that reported in Table 6. The algal biomass was harvested at the end of the exponential phase. In order to remove the biomass from the culture medium, microalgae were centrifuged at 5000 rpm for 10 min in a centrifuge JA 14. The obtained biomass was stored at −20 °C, and the amounts of AGS and HGS obtained were 5.20 and 4.80 g L^−1^ of wet biomass and 1.50 and 1.43 g L^−1^ of dry biomass, respectively.

### 4.3. Scanning Electron Microscopy

Dried samples of AGS and HGS were analyzed by scanning electron microscopy (SEM) using the ThermoFisher microscope model Phenom Pro Desktop SEM, having an electron optical magnification range: 80–150,000x; a resolution < 10 nm (BSED) and < 8 nm (SED); digital zoom: max 12x; acceleration voltages: default 5 kV, 10 kV, and 15 kV; vacuum modes: charge reduction mode (low vacuum mode)—high vacuum mode; and detector: BSD. 

### 4.4. Lipid Extraction

The microalgal biomass was lyophilized at −86°C, using a freeze-dryer (Lyovapor L200 Buchi) according to Lee et al. [57]. Total lipids were extracted from 1.0 g of dried biomass using a mixture of chloroform/methanol (2:1 v/v) according to Bligh et al. [58]. The FAMEs naturally present in the microalgae (methyl linoleate, methyl palmitate, hexadecanoic acid, methylester) were obtained. The fatty acid methyl esters naturally present in the microalgae were extracted using Soxhlet extraction, without any previous transmethylation, and were analyzed by GC-MS. The Soxhlet extraction was implemented with 2 g of sample powder on a Soxhtec system HT (Foss Soxtec 1043) for 6 h of extraction process at 140 °C, using hexane as solvent, followed by 30 min solvent rinse and 30 min solvent evaporation until the exhaustion of the oil contained in the microalgae. Only after Soxhlet extraction were the total lipids transmethylated to yield their corresponding fatty acid esters (FAMEs) using 2 mL of 1% NaOH in MeOH, followed by heating at 55 °C for 15 min at 55 °C. Next, 4.0 mL of 5% methanolic HCl were added and again heated for 15 min at 55 °C [59]. Finally, total FAMEs were eluted by adding 2.0 mL of *n*-hexane to the reaction mixture described above. The total FAMEs obtained were readily analyzed by GC-MS in order to determine the total saturated, monounsaturated and polyunsaturated fatty acids.

### 4.5. GC-MS Analysis

The *n*-hexane extracts were analyzed by GC-MS on an Agilent Technologies unit mod 6850—Series II, equipped with an auto sampler G45134 and an Agilent capillary column (DB-5 type, 0.18 mm ID, film 0.18 µm, length 20 m), using the Agilent Mass Selective Detector mod 5973. Helium was used as a carrier gas at a flow rate of 13.8 mL/min. The split ratio applied was 10:1. The injector temperature was 270 °C. The gradient applied was as follows: an isotherm of 2 min at 60 °C, a first ramp from 60 to 250 °C for 20 min (9.5 °C/min), followed by a second ramp from 250 to 300 °C for 10 min (10 °C/min). The temperature was then maintained at 300 °C for 5 min. All the analyses were carried out in triplicate, a confidential interval of 95% and a coverage factor K = 2 were applied. The limit of detection by GC-MS was 1 *p*mole per injection. In each case, the peak area was plotted against the standards concentration to obtain a linear relationship. As standard, a 37 component fatty acid methyl ester (FAME) mixture purchased from Supelco (37 Component FAME Mix Supelco Inc., Bellefonte, PA, USA) was used. Ergosterol (95% pure, GC assay), phytol (97% pure, GC assay), *n*-heptadecene (98% pure, GC assay), and nevronic acid (99% pure, GC assay) were purchased from Sigma (Sigma Aldrich, St. Louis, MO, USA). All the compounds utilized were analytical grade. Serial standard dilutions with hexane were made in triplicate to obtain concentrations of 15.000, 10.000, 5.000, 2.000, 1.000, 500, 200, 100, 50, and 25 μg/mL. A 1% lauric acid methyl ester (LAME, C12:0, Sigma-Aldrich) in hexane was prepared, and LAME equivalent to 5% of the total compounds was added to each dilution as an internal standard. The standard, the sample, and the internal standard solution as the compounds determination were carried out according to Lall et al. [60]. The identification of all the compounds was carried out by the interpretations of the mass spectra, in particular the analysis of fragment ions obtained, using the Nist Mass Spectral Library Program—version 2.0 software. The peak area of standards was plotted against the standard concentration to obtain a linear relationship. In particular, the coefficient of determination *r^2^* values obtained from the calibration curves were in the range between 0.98 and 0.99. Values of *r^2^* smaller than 0.98 were not accepted. Standard curves were in the same conditions of the sample analysis previously described. In each case, the peak area was plotted against the concentration to obtain a linear relationship. Specifically, the limit of detection (LOD), limit of quantification (LOQ), and *r^2^* values for each peak are reported in Table 7.

### 4.6. ATR-FTIR Analysis

Samples of AGS, HGS, and Sp were lyophilized and analyzed without any previous treatment and placed directly on the germanium piece of the infrared spectrometer with constant pressure applied (70 ± 2 psi). The FTIR spectra were recorded in the mid-IR region (4000–650 cm^−1^) at resolutions of 4 cm^−1^ with 32 scans using the Perkin Elmer FTIR Frontier coupled with DTGS (deuterated tri-glycine sulfate) detector (Perkin-Elmer Inc., Norwalk, CT, USA). Air background spectra was recorded and subtracted before analysis. To test repeatability, analyses were performed in triplicate and average spectra were used. Five samples for each group were analyzed. Spectra were baseline corrected and normalized, then elaborated using Spectrum Assure ID software, purchased with the instrument.

### 4.7. Statistical Analysis

The parametric test of one-way analysis of variance (ANOVA) after confirmation of normality and homogeneity of variance was used. Significant differences between experimental groups were evaluated by Duncan’s multiple range test. Significant differences were determined at the 0.05 level. Data were expressed as mean ± standard error of mean. The analyses were carried out with the Statistica version 7.0 statistical package (Statsoft Inc., Tulsa, OK, USA).

FTIR spectra were analyzed by the Spectrum AssureID software (trademark of PerkinElmer, Inc. part number 0993 4516 Release E; publication fate July 2006; Software Version 4.x). Assure ID employs the SIMCA algorithm (soft independent modeling class algorithm). Three classes were defined: AGS, HGS, and Sp. For cluster analysis, the spectral ranges (I) 3600–3000, (II) 2999–2800, (III) 1772–1712, (IV) 1711–1576, (V) 1575–1478, (VI) 1475–1175, (VII) 1174–950, and (VIII) 949–650 cm^−1^ were independently analyzed. Interclass distance between groups, recognition, and rejection rates of the samples were determined to evaluate the performance of the SIMCA model. 

Second derivative was employed to obtain more specific identification of little and very close absorption peaks, which were not well-resolved in the original spectrum. According to the Beer–Lambert law, absorbance is expressed as follows:(1)$$A\left(ῡ\right)=\alpha \left(ῡ\right)lc$$
where A is the wavenumber **ῡ** -dependent absorbance, *α* is the wavenumber-dependent absorption coefficient, *l* is the optical pathlength (mainly determined by the section thickness), and *c* is the concentration. When Equation (1) is differentiated twice, the result is
(2)$$\frac{d^{2}A\left(ῡ\right)}{d{ῡ}^{2}}=\frac{d^{2}\alpha \left(ῡ\right)}{d{ῡ}^{2}\;}\;lc$$

From Equation (2) it can be seen that quantitative information [61,62,63] can be obtained also from the second derivative spectra, as *l* and *c* are constant terms and are not affected by the differentiation. 

## 5. Conclusions

The present study reports how it is possible to obtain different biomolecules from *G. sulphuraria* microalga by changing the culture conditions that influence the metabolic processes. This outcome expands our knowledge about the microalgae metabolism, and presents innovative strategies for developing biotechnological applications. In particular *G. sulphuraria*, due to its interchangeable and versatile metabolism, appears to be a very good candidate for the co-cultivation with fungi or other beneficial microbes for the production of bioactive molecules useful for purifying wastewater, generating biomass that represents a renewable and sustainable feedstock for biofuel, nutraceutical, pharmacological, food, or feed production [64]. Although there are still more investigations required regarding microalgae metabolic changes, our data can have significant repercussions for potential biotechnological applications in the food, animal feed, nutraceutical, pharmacological, and energy fields.

## Figures and Tables

**Figure 1 marinedrugs-18-00169-f001:**
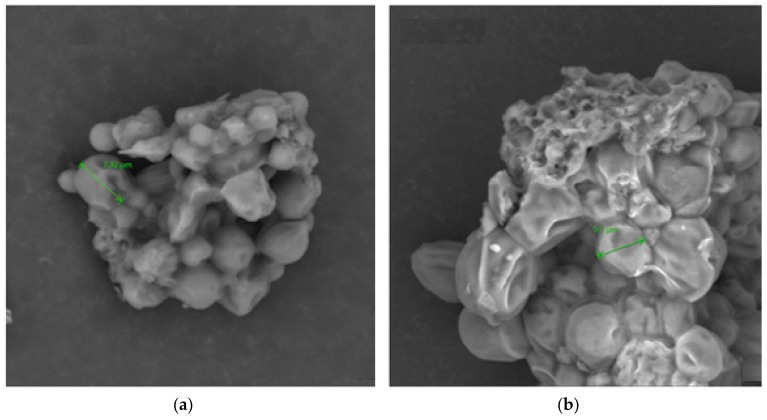
*Galdieria sulphuraria* strain ACUF 064 cultured in **(a)** heterotrophic (FOV: 62.5 µm, mode: 15kV-point, detector: BSD full) and **(b)** autotrophic conditions (FOV 39.5 µm, mode: 15kV-point, detector: BSD full).

**Figure 2 marinedrugs-18-00169-f002:**
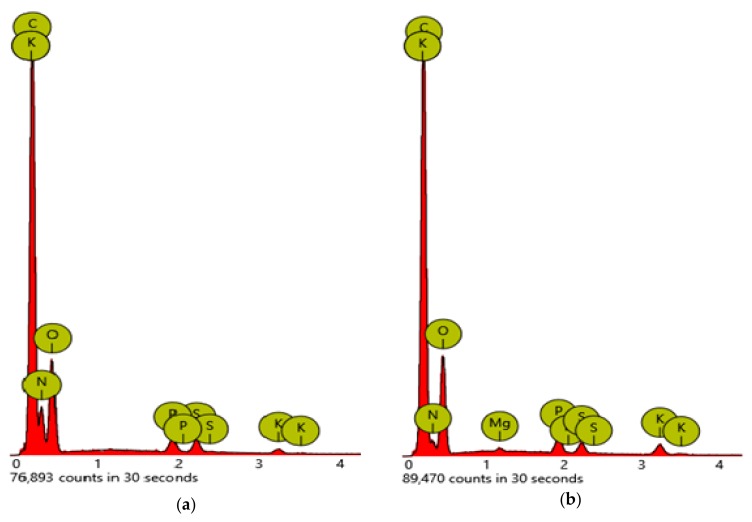
Different content of elements in *Galdieria sulphuraria* strain ACUF 064 cultured in heterotrophic **(a)** and autotrophic **(b)** conditions. Percentages are reported in Table 1.

**Figure 3 marinedrugs-18-00169-f003:**
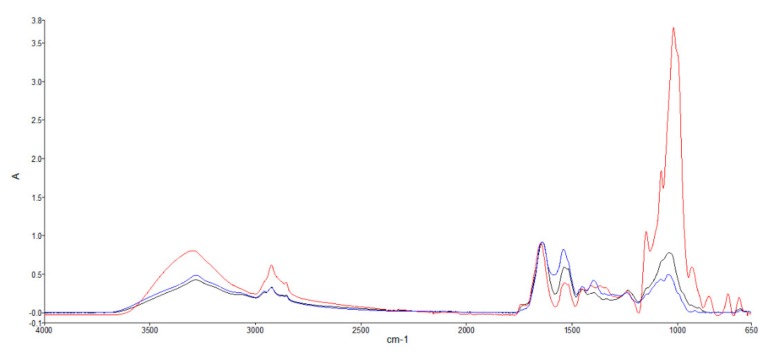
Infrared spectrophotometry (ATR-FTIR) spectra of *Galdieria sulphuraria* strain ACUF 064 cultured in autotrophic (**^____^**) and heterotrophic (**^___^**) conditions. (**^___^**) *Spirulina platensis*.

**Figure 4 marinedrugs-18-00169-f004:**
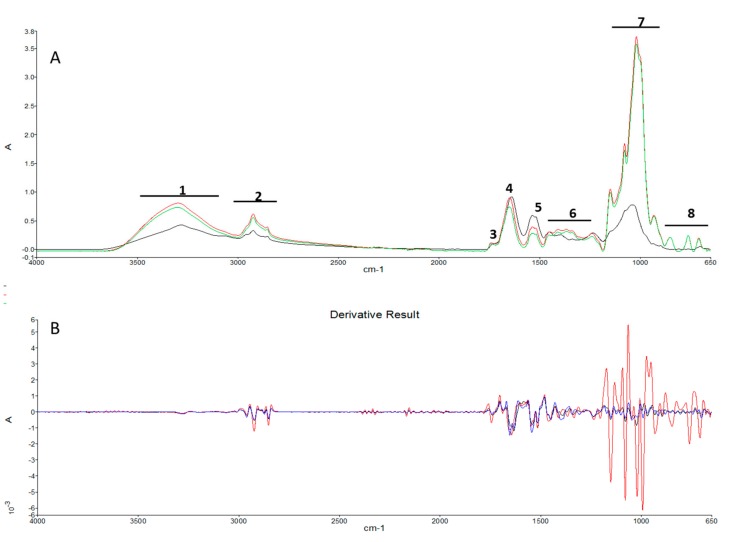
**(A)** Representative ATR-FTIR spectra of *Galdieria sulphuraria* strain ACUF 064 cultured in autotrophic (**^___^**) and heterotrophic (**^___^**) conditions and the substraction spectrum (**^___^**). **(B)** Second derivatives of *Galdieria sulphuraria* strain ACUF 064 cultured in autotrophic (**^___^**) and heterotrophic (**^___^**) conditions. (**^___^**) *Spirulina platensis*.

**Figure 5 marinedrugs-18-00169-f005:**
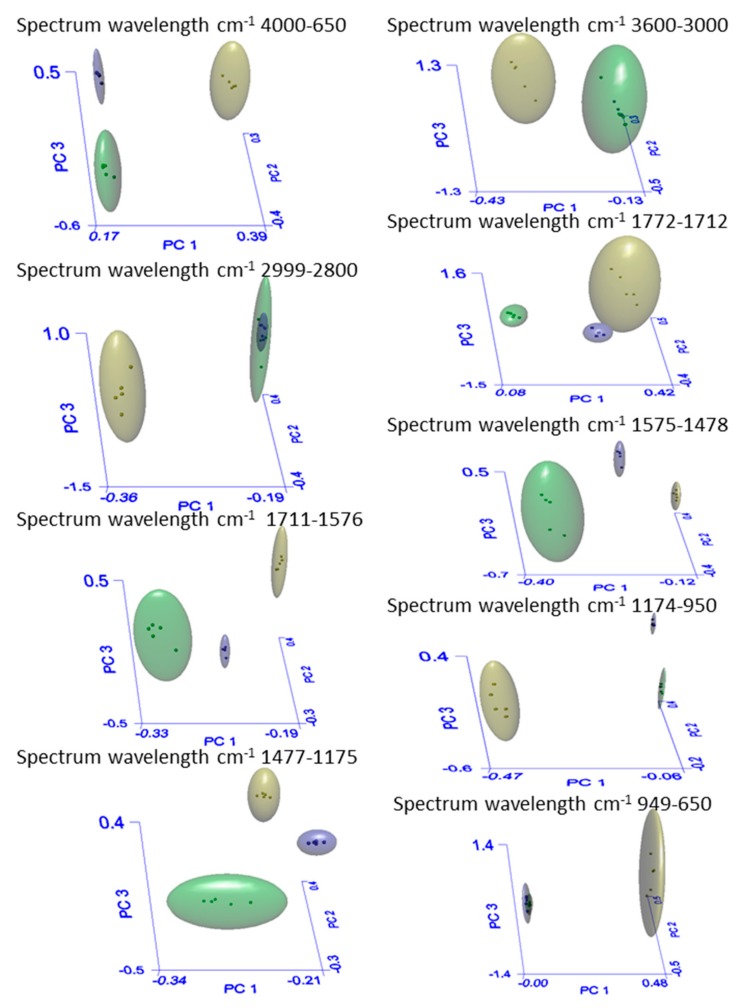
Three-dimensional principal component analysis score plot of *Galdieria sulphuraria* strain ACUF 064 cultivated in autotrophic (^▄^) and heterotrophic conditions (^▄^), plus *Spirulina platensis* in autotrophic conditions (^▄^). Data analysis was performed in the spectrum ranges reported in the rectangles above each plot.

**Table 1 marinedrugs-18-00169-t001:** Area values of different elements of *Galdieria sulphuraria* strain ACUF 064 cultured in autotrophic (AGS) and heterotrophic (HGS) conditions.

Element Number	Element Symbol	Element Name	Atomic Conc. HGS	Weight Conc. HGS (%)	Atomic Conc. AGS	Weight Conc. AGS (%)
6	C	Carbon	63.81	57.16	51.79	46.03
8	O	Oxygen	23.74	28.33	21.20	25.11
7	N	Nitrogen	11.40	11.90	26.40	27.37
15	P	Phosphorus	0.38	0.89	0.24	0.55
19	K	Potassium	0.32	0.93	0.11	0.31
16	S	Sulfur	0.27	0.64	0.26	0.62
12	Mg	Magnesium	0.08	0.15	0.00	0.00

**Table 2 marinedrugs-18-00169-t002:** Comparison of the fatty acid composition of *G. sulphuraria* strain ACUF 064 cultivated in autotrophic (AGS) and heterotrophic (HGS) conditions, and to *Spirulina platensis* (*Sp*) grown in autotrophic conditions. Values are reported as mean values (*n* = 3) ± SD, where SD is the standard deviation.

Molecular Formula	Peak	RT (min)	Compound	*AGS*	*HGS*	*Sp*
**C8:0**	1	7.53	Caprylic acid C8:0	0.060 ± 0.01	-	0.04 ± 0.03
**C13:0**	2	10.26	Tridecanoic acid	0.35+0.01	-	0.50 ± 0.02
**C14:0**	3	11.28	Myristic acid C14:0	1.74 ± 0.14^a^	1.90 ± 0.12^a^	0.13 ± 0.01^b^
**C14:1**	4	12.41	Myristoleic acid C14:1	0.10 ± 0.03	-	0.05 ± 0.04
**C15:0**	5	12.54	Pentadecanoic acid C15:0	0.61 ± 0.09 ^a^	0.36 ± 0.09 ^a^	0.03 ± 0.01^b^
**C16:0**	6	14.28	Palmitic acid C16:0	27.19 ± 0.12^b^	21.15 ± 0.31c	22.51 ± 0.27 ^a^
**C16:1**	7	15.96	Palmitoleic acid C16:1	0.32 ± 0.09^b^	0.33 ± 0.16^b^	4.74 ± 0.41 ^a^
**C17:0**	8	16.50	Heptadecanoic acid C17:0	0.27 ± 0.06 ^a^	0.31 ± 0.07 ^a^	0.16 ± 0.08^ab^
**C17:1**	9	18.62	*cis*-10-Heptadecenoic acid	0.26 ± 0.02^ab^	0.21 ± 0.08^b^	0.32 ± 0.04 ^a^
**C18:0**	10	19.43	Stearic acid	1.04 ± 0.11^b^	2.96 ± 0.06 ^a^	0.72 ± 0.11^c^
**C18:1 n9t**	11	21.07	Elaidic acid	0.15 ± 0.08 ^a^	0.17 ± 0.01 ^a^	0.04 ± 0.01^b^
**C18:1 n9c**	12	21.82	Oleic acid	20.91 ± 0.14^b^	30.07 ± 0.16 ^a^	2.95 ± 0.09^c^
**C18:3 n3**	13	24.01	Linolenic acid	5.90 ± 0.27 ^a^	3.31 ± 0.18^ab^	0.10 ± 0.03^c^
**C18:3 n6**	14	25.58	γ-Linolenic acid	-	-	13.15 ± 0.09
**C18:2 n6c**	15	26.13	Linoleic acid	18.91 ± 0.13 ^a^	14.31 ± 0.62^ab^	19.06 ± 0.51 ^a^
**C20:0**	16	28.25	Arachidic acid	0.05 ± 0.01	0,10 ± 0.07	0.04 ± 0.01
**C28H44O**	17	28.47	Ergosterol	-	10.21 ± 0.13^a^	2.93 ± 0.21^b^
**C20H40O**	18	29.75	Phytol	15.34 ± 0.14^b^	6.05 ± 0.09^c^	16.07 ± 0.76^a^
**C15H13N**	19	30.01	4’methyl-2-phenylindole	-	7.01 ± 0.03^a^	2.86 ± 0.04^b^
**C17H36**	20	33.47	n-Heptadecene	5.72 ± 0.35^b^	-	12.92 ± 0.47^a^
**C20:1**	21	33.61	cis-11-Eicosenoic acid	0.26 ± 0.11^b^	0.53 ± 0.02^a^	0.01 ± 0.01^c^
**C20:2**	22	34.08	cis-11,14-Eicosadienoic	0.57 ± 0.08 ^a^	0.65 ± 0.03^a^	0.25 ± 0.16^b^
**C20:3 n6**	23	34.12	cis-8,11,14-Eicosatrienoic acid	-	-	0.28 ± 0.07
**C20:3 n3**	24	35.03	cis-11,14,17- Eicosatrienoic acid	0.14 ± 0.05	0.28 ± 0.01	-
**C24:1**	25	35.97	Nervonic acid	0.11 ± 0.02	0.09 ± 0.01^ab^	0.14 ± 0.08^a^
**C19H34O2**	N.P.A.		Methyl linoleate	07.85 ± 0.16^a^	3.47 ± 0.03^b^	-
**C17H34O2**			Methyl palmytate	11.41 ± 0.73^a^	6.21 ± 0.03^b^	4.01 ± 0.62^b^
**C16H32O2**			Hexadecanoic acid, methyl ester	9.47 ± 0.49^a^	-	6.23 ± 0.31^b^
**∑-FATTY ACIDS**			**∑-FAME**	28.73 ± 0.74^a^	9.68 ± 0.03^b^	-
			**∑-SFA**	34.10 ± 0.21^b^	31.56 ± 0.03^c^	40.02 ± 0.26^a^
			**∑-MUFA**	30.11 ± 0.47^b^	38.54 ± 0.03^a^	8.25 ± 0.07^c^
			**∑-PUFA**	31.52 ± 0.83^b^	27.43 ± 0.61^c^	35.82 ± 0.62^a^

Organic compounds expressed as mean percentages of 100 mg of dry tissue weight. Values with different letters are significantly different (*p* < 0.05). N.P.A: naturally present in alga. See the Abbreviation section for the definitions of SFA, MUFA, and PUFA.

**Table 3 marinedrugs-18-00169-t003:** Peak assignment of *Galdieria sulphuraria* strain ACUF 064 cultured in autotrophic (AGS) and heterotrophic (HGS) condition and *S. platensis*, based on the literature [26,27].

Spectral Ranges Analyzed with SIMCA	Peak Wavelength (cm^-1^)			Peak Assignment	Macromolecules
	AGS	HGS	Sp		
3600–3000		3298		*v*(N-H) stretching of amide A	Proteins
	3284		3282		
2999–2800			2959	*v*as(CH2) and *v*s(CH2) stretching	Lipids, triglycerides, fatty acids, carbohydrates
	2924	2924	2925		
	2854	2855			
1772–1712		1743		*v*(C=O) stretching of esters	Cellulose–fatty acids
1711–1576	1640	1646	1641	Amide I *v*(C=O) stretching	Proteins
1575–1478	1538	1537	1541	Amide II *δ*(N-H) bending and *v*(C-N) stretching	Proteins
1477–1175	1453	1453	1452	*δ*as(CH2) and *δ*as(CH3) bending of methyl	Proteins, lipids
	1394	1411	1399	*δ*s(CH2) and *δ*s(CH3) bending of methyl; *v*s(C-O) of COO- groups; *δ*s(N(CH3)3) bending of methyl	Proteins and lipids
		1368			
	1336				
			1308	Amide III	Proteins
	1236	1238	1240	*V*as (>P=O) stretching of phosphodiesters	Nucleic acids and phospholipids
1174–950		1148		*v*(C-O-C)	Carbohydrates (including glucose, fructose, glycogen, etc.), polysaccharides
		1077	1079		
	1039		1043		
		1018			
949–650	806	931	916	Fingerprint region	
	763	850	880		
	700	760	743		
		662			

**Table 4 marinedrugs-18-00169-t004:** Representative peak area relative to the second derivative subtraction spectrum between *Galdieria sulphuraria* grown in heterotrophic conditions and autotrophic conditions. In the first column, the FT-IR ranges are reported, as shown in Figure 4a. The subtraction area (∆*_HGS–AGS_*) for each interval is expressed as the percentage of log_10_/total area.

*Spectral Ranges (cm^-1^)*			
FTr	Start	End	∆_HGS-AGS_
1	3600	3000	2.23 (18.63%)
2	2999	2800	1.16 (9.69%)
3	1772	1712	0.20 (1.67%)
4	1711	1576	1.62 (13.53%)
5	1575	1478	1.09 (9.11%)
6	1477	1175	1.78 (14.87%)
7	1174	950	2.60 (21.72%)
8	949	650	1.29 (10.78%)

**Table 5 marinedrugs-18-00169-t005:** Interclass distance, and recognition and rejection rates of *Galdieria sulphuraria* strain ACUF 064 cultivated in autotrophic conditions (AGS), heterotrophic conditions (HGS), and *Spirulina platensis* (Sp).

**Spectrum Wavelength cm** **^−1^ 4000–650**				
**Groups**	Recognition (%)^a^	Rejection (%)^b^	Interclass Distance^c^	
AGS	100(5/5)	100(10/10)	AGS–HGS	26.2
HGS	100(5/5)	100(10/10)	HGS-Sp	36.3
Sp	100(5/5)	100(10/10)	Sp-AGS	12.2
**Spectrum Wavelength cm** **^−1^ 3600–3000**				
**Groups**	Recognition (%)^a^	Rejection (%)^b^	Interclass Distance^c^	
AGS	100(5/5)	100(10/10)	AGS–HGS	21.9
HGS	100(5/5)	100(10/10)	HGS-Sp	21.2
Sp	100(5/5)	100(10/10)	Sp-AGS	6.56
**Spectrum Wavelength cm** **^−1^ 2999–2800**				
**Groups**	Recognition (%)^a^	Rejection (%)^b^	Interclass Distance^c^	
AGS	100(5/5)	100(10/10)	AGS–HGS	24.8
HGS	100(5/5)	100(10/10)	HGS-Sp	23.9
Sp	100(5/5)	100(10/10)	Sp-AGS	4.94
**Spectrum Wavelength cm** **^−1^ 1772–1712**				
**Groups**	Recognition (%)^a^	Rejection (%)^b^	Interclass Distance^c^	
AGS	100(5/5)	100(10/10)	AGS–HGS	15.3
HGS	100(5/5)	100(10/10)	HGS-Sp	14.2
Sp	100(5/5)	100(10/10)	Sp-AGS	6.5
**Spectrum Wavelength cm** **^−1^ 1711–1576**				
**Groups**	Recognition (%)^a^	Rejection (%)^b^	Interclass Distance^c^	
AGS	100(5/5)	100(10/10)	AGS–HGS	15.4
HGS	100(5/5)	100(10/10)	HGS-Sp	36.5
Sp	100(5/5)	100(10/10)	Sp-AGS	20.8
**Spectrum Wavelength cm** **^−1^ 1575–1478**				
**Groups**	Recognition (%)^a^	Rejection (%)^b^	Interclass Distance^c^	
AGS	100(5/5)	100(10/10)	AGS–HGS	15.5
HGS	100(5/5)	100(10/10)	HGS-Sp	24.2
Sp	100(5/5)	100(10/10)	Sp-AGS	21.7
**Spectrum Wavelength cm** **^−1^ 1477–1175**				
**Groups**	Recognition (%)^a^	Rejection (%)^b^	Interclass Distance^c^	
AGS	100(5/5)	100(10/10)	AGS–HGS	22.5
HGS	100(5/5)	100(10/10)	HGS-Sp	14.7
Sp	100(5/5)	100(10/10)	Sp-AGS	17.7
**Spectrum Wavelength cm** **^−1^ 1174–950**				
**Groups**	Recognition (%)^a^	Rejection (%)^b^	Interclass Distance^c^	
AGS	100(5/5)	100(10/10)	AGS–HGS	78.3
HGS	100(5/5)	100(10/10)	HGS-Sp	88.2
Sp	100(5/5)	100(10/10)	Sp-AGS	13.9
**Spectrum Wavelength cm** **^−1^ 949–650**				
**Groups**	Recognition (%)^a^	Rejection (%)^b^	Interclass Distance^c^	
AGS	100(5/5)	100(10/10)	AGS–HGS	26.7
HGS	100(5/5)	100(10/10)	HGS-Sp	27.9
Sp	100(5/5)	100(10/10)	Sp-AGS	6.22

Notes: ^a^ Percentage of recognition in optimal model should be closer to 100% ; ^b^ percentage of rejection in optimal model should be closer to 100% ; ^c^ interclass distances (ID) should be as high as possible ( minimum 3).

**Table 6 marinedrugs-18-00169-t006:** Composition of modified Allen medium (pH 1.5).

Components	g/L	Oligoelements	g/L
NaNO_3_	1.7	MnCl_2_ ∙4H_2_O	0.02
MgSO_4_∙7H_2_O	0.3	CuSO_4_∙5H_2_O	0.0001
K_2_HPO_4_	0.6	CoCl_2_∙H_2_O	0.00005
KH_2_PO_4_	0.3	Na_2_MoO_4_∙2H_2_O	0.00005
CaCl_2_∙2H_2_O	0.02	ZnCl_2_	0.00014
NaCl	0.05	H_2_SO_4_	0.30
FeSO_4_∙7H_2_O	0.1		

**Table 7 marinedrugs-18-00169-t007:** Limit of detection (LOD), limit of quantification (LOQ), and coefficient of determination (*r^2^*).

*Area*				*Height*		
Peak	LOD (ng/mL)	LOQ (ng/mL)	*r^2^*	LOD (ng/mL)	LOQ (ng/mL)	*r^2^*
**1**	0.21	0.63	0.9994	0.36	1.11	0.9987
**2**	0.19	0.57	0.9978	0.26	0.86	0.9986
**3**	0.30	0.90	0.9819	0.18	0.62	0.9956
**4**	0.14	0.42	0.9973	0.24	0.79	0.9996
**5**	0.15	0.46	0.9983	0.23	0.78	0.9983
**6**	0.19	0.58	0.9967	0.65	2,08	0.9972
**7**	0.20	0.61	0.9978	0.47	1.50	0.9998
**8**	0.33	0.97	0.9977	0.40	1.33	0.9996
**9**	0.22	0.68	0.9972	0.46	1.35	0.9988
**10**	0.19	0.59	0.9951	0.31	1.01	0.9986
**11**	0.14	0.43	0.9894	0.43	1.27	0.9991
**12**	0.16	0.47	0.9978	0.24	0.83	0.9980
**13**	0.21	0.63	0.9965	0.27	0.85	0.9899
**14**	0.23	0.69	0.9994	0.37	1.16	0.9996
**15**	0.18	0.56	0.9989	0.72	2.36	0.9881
**16**	0.16	0.48	0.9976	0.23	0.75	0.9893
**17**	0.22	0.70	0.9995	0.41	1.38	0.9957
**18**	0.21	0.63	0.9945	0.43	1.43	0.9995
**19**	0.24	0.73	0.9971	0.37	1.25	0.9992
**20**	0.27	0.81	0.9996	0.27	0.83	0.9948
**21**	0.21	0.64	0.9961	0.38	1.24	0.9996
**22**	0.25	0.76	0.9897	0.34	1.11	0.9982
**23**	0.18	0.54	0.9979	0.41	1.35	0.9993
**24**	0.17	0.50	0.9987	0.43	1.39	0.9975
**25**	0.19	0.59	0.9919	0.56	1.85	0.9967

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
