# Peer review of "Autotrophic and Heterotrophic Growth Conditions Modify Biomolecole Production in the Microalga Galdieria sulphuraria (Cyanidiophyceae, Rhodophyta)"

_marinedrugs, 2020, doi:10.3390/md18030169_

Round 1
Reviewer 1 Report
The manuscript (marinedrugs-743725), titled “Effect of different metabolic conditions on biomolecole production in the ancient extremophilic microalga Galdieria sulphuraria (Cyanidiophyceae Rhodophyta)” is interesting but several issues prevent its publication in Marine Drugs, at least in its current form.
A detailed revision on formatting and English should be done, starting on the title. The list of abbreviations is incomplete. The compounds designations such as “cis” and “n”, as well as the species names should be in italic.
Major concerns, the GC-MS analysis is very incomplete. Retention times, injection of standards in the same conditions, detailed information about the identification and quantification of the compounds, calibration curves (r2 from 98to 99????), LOD and LOQ values….
In order to say that bioactive compounds are produced, we need to be sure of its identification and quantification.
Author Response
Response to Reviewer 1 Comments
Point 1:
The manuscript (marinedrugs-743725), titled “Effect of different metabolic conditions on biomolecole production in the ancient extremophilic microalga Galdieria sulphuraria (Cyanidiophyceae Rhodophyta)” a detailed revision on formatting and English should be done, starting on the title.
Response 1:
The manuscript Please provide your response for Point 1. (in red)
The authors thank the Reviewer for the comments. The text has been revised both in terms of style format as well as the English language usage as suggested by the Reviewer.
The title has been modified to:
Autotrophic and heterotrophic growth conditions modify biomolecole production in the microalga Galdieria sulphuraria (Cyanidiophyceae Rhodophyta).
Point 2
The list of abbreviations is incomplete.
Response 2:
The list of abbreviations, as noted, has been completed. A check has been conducted by cross-checking the abbreviations reported in the manuscript and inserting all the used terms in the list.
Point 3
The compounds designations such as “cis” and “n”, as well as the species names should be in italic.
Response 3:
The designations “cis”, “n” and the species names, as suggested, were written in italics.
Point 4
Major concerns, the GC-MS analysis is very incomplete. Retention times, injection of standards in the same conditions, detailed information about the identification and quantification of the compounds, calibration curves (r2 from 98to 99????), LOD and LOQ values….In order to say that bioactive compounds are produced, we need to be sure of its identification and quantification.
Response 4:
As requested, all of the presented GC-MS data have been revised. In particular, the information, as indicated by the reviewer above, have now been inserted in the manuscript, in particular: conditions of use of the standards, identification and quantity of the compounds, calibration curves, detailed values of LOD, LOQ and r2.
Reviewer 2 Report
line 27: Delete "organisms that are"
line 27: Replace "togheter" with "together"
line 28: Remove comma
line 28: Replace "do not share" with "not sharing"
line 29: Replace "the metabolism, in fact" with "their metabolisms. In fact,"
line 29: Replace "the first" with "algae"
line 29: Replace "chrolophyll" with "chlorophyll"
line 30: Insert a comma after "organisms"
line 30: Delete ",on the contrary,"
line 49: Replace "waters, they" with "waters. They"
line 50: Replace "Tallophytes however" with "Tallophytes. However,"
line 51: Delete "on the contrary of fungi"
line 52: Replace "potentiality" with "potential"
line 85: Place brackets around reference number 15, i. e. [15]
lines 111, 112, 218, 402, 403, 404, 405: Use italics on species names, i. e. G. sulphuraria or Galderia sulphuraria, Spirulina platensis
line 13: Table 2, not Table 4.
**Very important: In Table 2, your mean percentages of compounds add up >100%, which is impossible. In the case of AGS, the sum is 124%, and in the case of HGS, it is 107%. Check your GC-MS data and correct this**
line 118: Place a period after "linoleate"
line 118: Delete "with respect to the heterotrophic conditions."
line 123: Delete comma after "autotrophic" and replace "eterotrophic" with "heterotrophic"
line 201: The "D" in "D-glucose" should be 10 pt.
Under Table 2 note, add "See Abbreviation section for the definitions of SFA, MUFA, and PUFA."
Table 2 and line 209: Change "methylester" to "methyl ester"
line 210: Replace "to avoid" with "the avoidance of"
line 213: Replace "alcohol, its" with "alcohol. Its"
line 214: Replace "PYL" with "PYT"
lines 229-232: Remove the bold and italics
line 243: Insert "to" before "promote"
line 243: Delete "the raising of"
line 245: Replace "has" with "have"
line 245: Replace "neither in the" with "either under"
line 246: Replace "nor in the" with "or"
line 252: Delete "in AGS and"
line 253: Replace "human health, in fact" with "human health. In fact,"
lines 254-255: Replace comma with period after "LDL-Cholesterol" and delete "known as ... cardiovascular system."
Table 6: Replace all "x" in chemical formulas with middots. Middots are used for hydrates.
line 301: Replace "m2" with "m2"
lines 303, 305, 314: Replace the symbol for liter with the correct international standard: "L" not "l"
line 326: Replace "extracted" with "obtained"
line 326: Replace "soxeleth" with "Soxhlet extraction"
line 327: Delete "subsequently"
line 327: Replace "reaction they were" with "and"
line 331: Insert "Soxhlet extraction were" after the word "after"
Author Response
Response to Reviewer 2 Comments
We thank the reviewer for the detailed corrections to the text. Please note that initially, all of the revisions as indicated by Reviewer 2 were addressed point by point below and in the text. However, subsequently, during the process of conducting a complete revision of the style format and the English usage in the manuscript (as requested during the review process) these specific modifications, as indicated, may no longer be apparent in the final version uploaded.
Point 1:
line 27: Delete "organisms that are"
line 27: Replace "togheter" with "together"
Response 1:
line 27: “organism that are” was deleted
line 27: has been replaced “togheter” with “together”
Point 2:
line 28: Remove comma
line 28: Replace "do not share" with "not sharing"
Response 2:
line 28: comma has been removed
line 28: “do not share” was been replaced with “not sharing”
Point 3:
line 29: Replace "the metabolism, in fact" with "their metabolisms. In fact,"
line 29: Replace "the first" with "algae"
line 29: Replace "chrolophyll" with "chlorophyll"
Response 3:
line 29: "the metabolism, in fact" has been replaced with "their metabolisms. In fact,"
line 29: "the first" has been replaced with "algae"
line 29: "chrolophyll" has been replaced with "chlorophyll"
Point 4:
line 30: Insert a comma after "organisms"
line 30: Delete ",on the contrary,"
Response 4:
line 30: after “organism” has been inserted a comma
line 30: ", on the contrary” has been deleted
Point 5:
line 49: Replace "waters, they" with "waters. They"
Response 5:
line 49: "waters, they" has been replaced with "waters. They"
Point 6:
line 50: Replace "Tallophytes however" with "Tallophytes. However,"
Response 6:
line 50: "Tallophytes however" has been replaced with "Tallophytes. However,"
Point 7:
line 51: Delete "on the contrary of fungi"
Response 7:
line 51: "on the contrary of fungi" has been deleted.
Point 8:
line 52: Replace "potentiality" with "potential"
Response 8:
line 52: "potentiality" has been replaced with "potential"
Point 9
line 85: Place brackets around reference number 15, i. e. [15]
Response 9:
has been place brackets around reference number 15, i. e. [15]
Point 10
lines 111, 112, 218, 402, 403, 404, 405: Use italics on species names, i. e. G. sulphuraria or Galderia sulphuraria, Spirulina platensis
Response 10:
lines 111, 112, 218, 402, 403, 404, 405: We have used, as suggested, italics on species
names, i. e. G. sulphuraria or Galderia sulphuraria, Spirulina platensis
Point 11
Line 13: Table 2, not Table 4.
Response 11:
Line 13 Table 4 has been replaced with Table 2.
Point 12
Very important: In Table 2, your mean percentages of compounds add up >100%, which is impossible. In the case of AGS, the sum is 124%, and in the case of HGS, it is 107%. Check your GC-MS data and correct this
Response 12:
In the Table 2, due to an editing error, absolute values have been reported instead of percentage values. As suggested, the previous data were replaced by reporting the percentage values of the compounds
Point 13
line 118: Place a period after "linoleate"
line 118: Delete "with respect to the heterotrophic conditions."
Response 13:
line 118: a period after “linoleate” was inserted
line118: "with respect to the heterotrophic conditions" has been deleted.
Point 14
line 123: Delete comma after "autotrophic" and replace "eterotrophic" with "heterotrophic"
Response 14:
comma after "autotrophic" was deleted and replaced with "eterotrophic"
Point 15
line 201: The "D" in "D-glucose" should be 10 pt.
Response 15:
The "D" in "D-glucose" now is 10 pt.
Point 16
Under Table 2 note, add "See Abbreviation section for the definitions of SFA, MUFA, and PUFA."
Response 16
Under Table 2 note, has been added "See Abbreviation section for the definitions of SFA, MUFA, and PUFA."
Point 17
Table 2 and line 209: Change "methylester" to "methyl ester"
Response 17
Table 2 and line 209: "methylester" has been replaced with "methyl ester"
Point 18
line 210: Replace "to avoid" with "the avoidance of"
Response 18
line 210: Has been replaced "to avoid" with "the avoidance of"
Point 19
line 213: Replace "alcohol, its" with "alcohol. Its"
Response 19
line 213: "alcohol, its" has been replaced with "alcohol. Its"
Point 20
line 214: Replace "PYL" with "PYT"
Response 20
line 214: "PYL" has been replaced with "PYT"
Point 21
lines 229-232: Remove the bold and italics
Response 21
lines 229-232: the bold and italics have been removed;
Point 22
line 243: Insert "to" before "promote"
line 243: Delete "the raising of"
Response 22
line 243: has been inserted "to" before "promote" and deleted "the raising of"
Point 23
line 245: Replace "has" with "have"
line 245: Replace "neither in the" with "either under"
Response 23
line 245: has been replace "has" with "have" and "neither in the" with "either under"
Point 24
line 246: Replace "nor in the" with "or"
Response 24
line 246: Has been replace "nor in the" with "or"
Point 25
line 252: Delete "in AGS and"
Response 25
line 252: Has been deleted "in AGS and".
Point 26
line 253: Replace "human health, in fact" with "human health. In fact,"
Response 26
line 253: Has been replaced "human health, in fact" with "human health. In fact,"
Point 27
lines 254-255: Replace comma with period after "LDL-Cholesterol" and delete "known as ... cardiovascular system."
Response 27
lines 254-255: Has been replaced comma with period after "LDL-Cholesterol" and deleted "known as ... cardiovascular system."
Point 28
Table 6: Replace all "x" in chemical formulas with middots. Middots are used for hydrates.
Response 28
Table 6: all "x" in chemical formulas has been replaced with middots.
Point 29
line 301: Replace "m2" with "m2"
Response 29
line 301: "m2" has been replaced with "m2"
Point 30
lines 303, 305, 314: Replace the symbol for liter with the correct international standard: "L" not "l"
Response 30
lines 303, 305, 314: the symbol for liter “l” has been replaced with the correct international standard: "L".
Point 31
line 326: Replace "extracted" with "obtained"
line 326: Replace "soxeleth" with "Soxhlet extraction"
Response 31
line 326: has been replaced "extracted" with "obtained" and "soxeleth" with "Soxhlet extraction"
Point 32
line 327: Delete "subsequently"
line 327: Replace "reaction they were" with "and"
Response 32
line 327: "subsequently" has been deleted and "reaction they were" was replaced with "and"
Point 33
line 331: Insert "Soxhlet extraction were" after the word "after"
Response 33
Line 331: "Soxhlet extraction were" has been inserted after the word “after”
Round 2
Reviewer 1 Report
The manuscript was greatly improved and is now suitable for publication.